# On the Use of Hair Analysis for Assessing Arsenic Intoxication

**DOI:** 10.3390/ijerph16060977

**Published:** 2019-03-18

**Authors:** Sidney A. Katz

**Affiliations:** Chemistry Faculty, Rutgers University, Camden, NJ 08102, USA; skatz@scarletmail.rutgers.edu

**Keywords:** hair analysis, arsenic intoxication, arsenic, ADME, arsenic methylation

## Abstract

Correlations between the concentrations of arsenic in scalp hair and in drinking water as well as in blood and/or urine have been reported. These correlations clearly show exposure–absorption–excretion relationships. In addition, arsenic metabolites such as monomethylarsonic acid and dimethylarsinic acid have been identified and quantified in these tissues and fluids, leaving little doubt that elevated levels of arsenic in the hair can reflect systemic arsenic intoxication. Consequently, hair analysis has potential merit as a screening procedure for poisoning by arsenic. However, questions regarding the exogenous versus the endogenous deposition of arsenic in the hair, and uncertainties about the normal level of arsenic in the hair remain unresolved. Pending their resolution, the determination of arsenic in hair should remain a screening tool, and clinical signs and symptoms should be employed to complete the diagnosis of arsenic poisoning.

## 1. Introduction

Since the middle of the 20th Century, the determination of trace element levels in human scalp hair has become increasingly popular for monitoring environmental exposures, evaluating heavy metal poisonings, assessing nutritional status, and diagnosing diseases. Blood and urine analysis are the more traditional approaches to testing trace element levels in the human body, but the bio-kinetics as well as changing external factors often lead to fluctuating trace element concentrations in blood and urine. Hair, on the other hand, is considered by some to be a metabolic end product providing a more permanent record of the trace elements associated with health and disease and of trace elements assimilated from the environment.

## 2. Biological Basis

In 1942, Schönheimer [1] described body tissues as being in a state of dynamic equilibrium, or homeostasis, which tended to maintain stability with coordinated responses to compensate for changes in the physiological system. Hair is an exception. Unlike other body compartments, hair is a metabolic end product that is thought to incorporate trace elements into its structure during its growing process. During the growth phase of a hair cycle, the matrix cells at the papilla of the hair follicle show intense metabolic activity and produce hair at a rate of approximately 0.3 mm/day. The developing hair is exposed to this metabolic milieu for only a short period of time, during which trace elements from the blood are partitioned into the nascent hair. As the growing hair approaches the skin surface, it undergoes a hardening process, or keratinization, and the trace elements accumulated during its formation are sealed into the protein structure of the hair. Flesch [2] hinted at the possibility of hair being a minor excretory organ for arsenic and possibly other toxic elements more than 60 years ago. It is on this basis that the trace element concentrations of hair are possibly related to the trace element concentrations in the body.

## 3. Drawbacks

The strongest argument against using hair mineral analysis as a measure of trace elements in the body is the difficulties encountered in differentiating between those deposited by metabolic activity and those present in/on the hair as a result of exogenous deposition. It is for this reason that many hair analysis protocols involve some sort of pre-cleaning to remove this external contamination prior to the measurement of the trace element concentrations. The lack of a standardized pre-treatment, however, has led to divergent results among laboratories using different protocols, thereby hindering the establishment of “normal values” for the trace elements in hair. These as well as other concerns were discussed by an ATSDR (Agency for Toxic Substances and Disease Registry) Panel in 2001 [3]. Among the factors identified by the panel as limiting the interpretation of even the most accurate, reliable, and reproducible laboratory results were:(1)The lack of reference ranges in which to frame the interpretation of results.(2)Difficulties in distinguishing between endogenous and exogenous deposition in hair.(3)A lack of understanding of how and to what extent environmental contaminants are incorporated into the hair.(4)A lack of correlation between levels in hair and blood and other target tissues, as well as a lack of epidemiological data linking substance-specific hair levels with adverse health effects.

These limitations are similar to those identified by Katz and Wood [4], who wrote some two decades earlier: “While many laboratories enthusiastically accept the use of scalp hair as a biopsy tissue, several serious questions remain. Before trace element levels in scalp hair can become an acceptable diagnostic indicator of mineral metabolism, additional work is required in four specific areas: (1) normal values showing narrow ranges must be established, (2) sample collection and sample preparation must be standardized, (3) trace elements incorporated into the hair from within the body (endogenous) must be differentiated from contamination by external (exogenous) sources, and (4) standard reference materials must become available to allow laboratories to establish quality assurance programs.”

These issues remain unresolved. Using time-of-flight secondary ion mass spectrometry, Kempson and Skinner [5] demonstrated the removal of calcium, potassium, and sodium from the surface of hair as well as from inside hair washed by two procedures. More recently, Pozebon and her co-workers [6] published a comprehensive review of elemental analysis in hair that contained an extensive tabulation of the washing procedures employed. Among the conclusions they made from their review was: “Hair can be an indicator of poisoning or contamination providing chronological information over months or years. Though, it is mandatory to distinguish endogenous from exogenous sources of investigated elements.”

## 4. Correlations with Exposure

Nonetheless, there is evidence confirming that hair arsenic concentrations reflect exposure to arsenic and its presence in other body compartments. Hindmarsh et al. [7] have reported positive correlations between the concentrations of arsenic in hair and in drinking water with the clinical signs of chronic arsenic poisoning. Their study group consisted of 92 individuals consuming water with arsenic concentrations in excess of 0.05 p.p.m. (the current US EPA, United States Environmental. Protection Agency, limit for arsenic in municipal water supplies is 0.01 p.p.m.) from domestic wells in Waverly, Nova Scotia. The work of Valentine and her coworkers [8] shows the efficiency of hair analysis for evaluating systemic arsenic intoxication. Samples of blood, urine, hair, and tap water were obtained from some 150 residents of two California and three Nevada communities where the ground water contained 6, 51, 98, 123, and 393 p.p.b. arsenic. The arsenic concentrations were determined by AAS (Atomic Absorption Spectrometry, CSIRO, Melbourne, Australia) of the gaseous hydride. The results showed “dose–response” relationships for regressions of the logarithms of the concentrations of arsenic in the urine and in the hair with the logarithms of the concentrations of arsenic in the tap water. A “dose–response” for the regression of the logarithms of the concentrations of arsenic in blood with the concentrations of arsenic in the tap water was not observed at tap water arsenic concentrations below 100 p.p.b. For this reason, it was concluded that, “… blood does not seem to be useful in assessing the degree of arsenic exposure. Hair and/or urine are useful…” A similar study [9] involving 50 residents of a Mexican village where the drinking water supply contained 0.41 p.p.m. arsenic and an equal number of control subjects whose water supply contained 5 p.p.b. arsenic had similar results and a similar conclusion. These results are reproduced in Table 1. The authors noted that the exposed subjects with the highest hair arsenic concentrations displayed the cutaneous signs of arsenic poisoning. Yang et al. [10] examined 311 subjects from a rural region in Inner Mongolia where the prevalence of arsenism was reported to be 15.53%. They found correlations between the arsenic concentrations in hair and the severity of dermal manifestations of arsenic intoxication and between the arsenic concentrations in hair and the arsenic concentrations in drinking water. Concha et al. [11] reported on urine and hair arsenic concentrations of female subjects exposed to arsenic from drinking water in eight regions of Argentina. The average arsenic concentrations in the water from these eight locations and the average concentrations of arsenic in the urine of the woman from these eight locations are summarized in Table 2. There was a significant correlation between the average concentration in drinking water and the average concentration in urine, r = 0.96, *p* < 0.001, but there were marked variations in the urinary arsenic concentrations among individuals from each location. In addition, there was a significant correlation between the average arsenic concentrations in the urine samples and the average arsenic concentrations in hair, r = 0.64, *p* < 0.001. As was the case with the drinking water correlation, there were variations in the hair arsenic concentrations for similar concentrations in urine. In one case where hair contained 1500 p.p.b. arsenic and the corresponding urine contained 64 p.p.b., the donor was suspected of bathing in the arsenic-rich water, with 6000 p.p.b., at the thermal spa in San Antonio de los Corbes. Two of the three investigators explored the possibility of exogenous contamination by measuring the arsenic contents in their own hair before and after washing it with the arsenic-rich water at San Antonio de los Corbes. The pre-washing arsenic concentrations were 33 and 78 p.p.b. The corresponding post-washing arsenic concentrations were 395 and 461 p.p.b. This observation clearly supports the external rather than the internal (i.e., metabolic) deposition of arsenic in the hair. The inability to distinguish between exogenous and endogenous deposition is one of the most compelling reasons to reject hair analysis as an indicator of the arsenic body burden. It is for this reason the authors concluded: “The arsenic concentration in urine seems to be a better marker of individual arsenic exposure than concentrations in drinking water and hair.” Marchiset-Ferlay et al. [12] have raised the question, “what is the best biomarker to assess arsenic exposure via drinking water?” Their review of the literature focused on five: Concentrations of arsenic in blood, concentrations of arsenic in urine, concentrations of arsenic in hair and nails, concentrations of porphyrins (many environmental toxins induce porphyrinurias. The heptacarboxy-, hexacarboxy- and pentacarboxyporphyrins are uniquely elevated by arsenic exposure) in blood and urine, and genotoxic effects such as DNA damage, sister chromatid exchange, chromosomal aberrations, and micronuclei assay. Of these five, they concluded the concentrations of arsenic in urine and toe nails were useful indicators of the internal arsenic concentrations. Liu et al., [13] reported saliva, urine, nails, and hair can be used as biomarkers of arsenic exposure. Others [14,15,16,17,18] have endorsed the use of hair analysis for assessing arsenic intoxication. Much of this support is due to the presence of arsenic metabolites in the hair.

## 5. Absorption

Arsenic intoxication can occur by ingestion, inhalation and, to a lesser extent, dermal absorption. Arsenic absorption from the gastrointestinal tract, from the respiratory tract, and through the skin depends on factors such as oxidation state, solubility, chemical form, et cetera [19,20,21,22]. Experimental data indicate that >90% of an ingested dose of dissolved inorganic trivalent arsenic, iAs(III), or of dissolved inorganic pentavalent arsenic, iAs(V), is absorbed from the gastrointestinal tract. Organo-arsenals, such as the arsenobetaine, [(CH_3_)_3_As]+[CH_2_COO]−, and trimethylarsonium acetate, in some sea foods, are also readily absorbed (~80%). The absorption of ingested arsenic from compounds of low solubility is less. Absorption of arsenic by the inhalation route is dependent upon solubility and particle size. As in the case with the absorption of arsenic by ingestion, absorption from inhaled compounds of low solubility is less. Absorption from inhaled particles smaller than 2 µm (MADD) is greater than absorption from larger particles because the latter are less able to penetrate into the alveoli. It has been estimated that about 40% of the arsenic in cigarette smoke was deposited in the lungs, and 80% of that was absorbed. Lazarević et al. [23] analyzed 80 brands of cigarettes. They found the arsenic contents of them ranged from <0.02 to 0.71 p.p.m. Percutaneous absorption of arsenic from aqueous solution was between 0.6 and 4.4% of the applied dose [24].

## 6. Distribution

Following absorption, blood is the main vehicle for the transport of arsenic. Postmortem examinations of the occupationally-exposed show distributions of arsenic to muscles, bones, kidneys, and lungs as well as to nails and hair. Organ concentrations (as total arsenic on a dry mass basis) from a person who committed suicide and who died 3 days after ingesting approximately 8 grams of arsenic trioxide, As2O_3_, are shown in Table 3 [25]. Table 4 [25] shows the distributions of the chemical forms of arsenic in these organs, possibly indicating detoxification by oxidation and methylation. Arsenic was methylated in the liver and excreted in the urine. Half time for the elimination of arsenic (administered as a radiolabeled pentavalent compound) was on the order of two days. On average, the proportions of methylated arsenic excreted in the urine were 70–80% DMA and 10–15% MMA. Some 10–25% of the urinary arsenic was not methylated.

## 7. Metabolism

As many as six different forms of arsenic may be involved in its methylation. These possibilities are shown in Table 5. Buchet et al. [26] administered sub-milligram quantities of arsenic as sodium arsenite (a trivalent compound) orally to four volunteers daily until a steady state of elimination was reached at five days. Analysis of the urine for total arsenic (tAs), inorganic arsenic (iAs), monomethylarsonic acid (MMA), and dimethylarsinic acid (DMA) showed tAs ~ = iAs + MMA + DMA, indicating iAs, MMA, and DMA were the only metabolic forms of arsenic excreted in the urine after exposure to arsenic by the oral route. Urinary excretion amounted to approximately 60% of the ingested dose. The biological half life of the ingested arsenic ranged from 39 to 59 hours depending on the amount (from 0.125 to 1000 mg) of arsenic ingested. Hayakawa et al. [27] investigated the role of arsenic–glutathione complexes (As–GHS) as substrates for methyltransferase (Cyt19) in the formation of MMA and DMA. The results of their analysis by HPLC–ICPMS suggested the CYT19 catalyzed the transfer of a methyl group from S-adenosyl-L-methionine to arsenic as the arsenic triglutathione (ATG) complex to form monomethylarsonic digulathione (MADG), which became the Cyt19 substrate for further methylation to dimethylarsenic glutathione (DMAG). Hydrolysis of MADG and DMAG and subsequent oxidation produced MMA and DMA which were excreted in the urine.

Thomas et al. [28] proposed “conceptual models for methylation of inorganic arsenic catalyzed by AS3MT. In model A, oxidative methylation of trivalent arsenicals alternates with reduction of pentavelent arsenicals. In this pathway, arsenite (AsIII) is converted to methylarsonic acid (MAsV), which is reduced to methylarsonous acid (MAsIII). This arsenical is converted to dimethylarsenic acid (DMAsV,), which is reduced to dimethylarsinous acid (DMAsIII). This dimethylated species is converted to trimethylarsine oxide (TMAsV), which is reduced to trimethyl arsine (TMAsIII). Model B involves the sequencial addition of methyl groups to trivalent arsenicals that are complexed to glutathione (GHS). In each model, S-adenosylmethionine (AdoMet) is the methyl group donor; S-adenosylhomocystine (AdoHcy) is produced by removal of a methyl group from AdoMet”.

## 8. Metabolites in Hair

DMA, MMA, and iAs were found both in the urine and in the hair of Japanese workers at a copper smelter and at a gallium arsenide production facility [29], confirming absorbed arsenic was metabolically deposited in the hair. Shraim et al. [30] reported on a methodology for the speciation of arsenic using scalp hair specimens from survivors of a poisoning incident in Wakayama and from the NIES CRM No. 13 (National Institute of Environmental Studies, Japan Certified Reference Material No. 13). Samples of hair were cryopulverized and shaken with water for 30 min to extract the arsenic. The aqueous extracts were separated by centrifugation, and the residuals were extracted twice more. The arsenic species in the extracts were separated and identified by HPLC. Quantification was by ICPMS. Their results for hair from two of the survivors and for the CRM are summarized in Table 6. The uncertified reference value for total arsenic in NIES CRM No. 13 is 100 p.p.b. [31]. It appears the efficiency of their extraction procedure was only 39%. Mandal et al. [32] also employed HPLC and ICPMS to identify and quantify the arsenic species in aqueous extracts of hair and fingernail specimens collected from 47 exposed subjects in West Bengal. They found most of the arsenic extracted from the hair of the West Bengal subjects was inorganic, iAs(III) and iAs(V). Dimethylarsinous acid was reported to be present in the extracts from the fingernails at concentrations exceeding those of MMA and DMA. Similarly, Yañez et al. [33] reported that more than 98% of the arsenic extracted from hair samples collected from Chilean villagers exposed to arsenic in their drinking water was inorganic. While the arsenic in the drinking water was mostly iAs(V), more iAs(III) than iAs(V) was found in the hair extracts. However, Raab and Feldmann [34] reported that, “… the pentavalent form was always the dominant form after extraction. Hair and nail samples from humans suffering from chronic arsenic intoxication contained dominantly inorganic arsenic with small and strongly varying amounts of DMA and MMA present.” Piñeiro et al. [35] evaluated pressurized hot water extraction for the speciation and quantification of arsenic in hair. Hair samples from unexposed volunteers were washed with water and acetone and pulverized in a vibrating zirconia ball mill prior to the pressurized hot water extraction. Speciation and quantification of arsenic in the extracts were by HPLC–ICP-MS. The major species found in the extracts were inorganic, iAs(III) and iAs(V). The Chinese analysis standard substance BGW07601 Human Hair Powder, the certified total arsenic content of which is 280 ± 5 p.p.b., was included among the samples. The results, as p.p.b., for the arsenic species in this reference material were: iAs(III) = 94.2 ± 2.2, iAs(V) = 130.6 ± 3.3, MMA < 28.7, DMA < 21.0, sum total 224.8 ± 3.8 p.p.b. MMA and DMA concentrations were below the LoQs (Limits of Quantification). Arsenic recovery was 80%. Chen et al. [18] identified nine forms of arsenic in aqueous extracts of most of the hair samples collected from patients undergoing treatment for promyelocytic leukemia with daily doses of 10 mg As_2_O_3_ iv. In addition to iAs(III), iAs(V), MMA and DMA, they found monomethylarsenous acid, monomethylmonothioarsonic acid, dimehtylmonothioarsinic acid, and two unidentified arsenic species using HPLC–ICPMS. Their results are summarized in Table 7.

There is little doubt that exposure to arsenic in domestic drinking and bathing water is reflected by the presence of arsenic in the hair. Experimental data for eight locations in Argentina [11] showed correlations between the concentrations of total arsenic in the drinking water and in the urine (r = 0.96, *p* < 0.001) and between the total arsenic concentration in urine and in hair (r = 0.064, *p* < 0.001). A positive correlation between the concentrations of arsenic in drinking water and in hair (r = 0.86, *p* < 0.0001) was also reported for Cambodian subjects [15]. Data collected from an exposed population in Bangladesh also showed a positive correlation for log–log transformed hair concentrations–water concentrations (r = 0.55, *p* < 0.001) [36]. Using data collected from residents of eight Chinese villages, Cui et al. [37] reported a correlation between the concentration of arsenic in the drinking water and in hair (r = 0.344, *p* < 0.05). In addition, a positive correlation between arsenic concentrations in the water and arsenic concentrations in the urine (r = 292, *p* < 0.05) was reported. The arsenic species shown in Table 5 were excreted in the urine and hair. The methylated species were of metabolic origin.

Almost without exception, inorganic arsenic, iAs, was the dominant form of arsenic found in human hair. Of the methylated species found in hair, the concentrations of DMA were greater than those of MMA. This appears to be the case for urine too, but the concentrations for the methylated species exceeded those of inorganic arsenic, iAs, in urine. The relative hair and urine concentrations of iAs(III), iAs(V), MMA and DMA normalized to DMA are summarized in Table 8. The original data were normalized because some were reported as percentages and others were reported as p.p.b. These data showed few, if any, relationships between the speciation of the excreted arsenic.

Human metabolism and excretion of arsenic appears to be age dependent. Chowdhury et al. [44] observed children excrete more arsenic per kg body mass than adults did, and urinary MMA was 6½ % lower for children than adults, while urinary DMA was 10% higher in children than adults. A similar observation was reported earlier by deCastro et al. [45]. Brima, et al. [46] have reported that human arsenic metabolism appears to depend upon ethnicity. They found statistically significant, *p* < 0.05, differences in the levels of total arsenic in urine and finger nail samples collected from Asian, Black (Somali African), and White residents of Leicester, U.K. Of the total arsenic in urine samples from Asian subjects, 16% was present as DMA compared to 50% for African subjects and 22% for White subjects. Mean values for the total arsenic concentrations in hair samples collected from subjects in the three groups (117, 116 and 141 p.p.b., respectively) did not differ significantly, with *p* > 0.05. The total arsenic concentrations in urine and finger nail samples from the African subjects were higher as was the percentage of urinary DMA. These observations were interpreted as being suggestive of a different pattern of arsenic metabolism in this ethnic group. In addition, different patterns of arsenic metabolism based on gender have been reported [47,48]. A mathematical relationship based on methylated arsenic species and arsenic in water, however, remains elusive.

## 9. Elimination

Absorbed arsenic is eliminated from the body as at least four chemical forms, iAs(III), iAs(V), MMA and DMA, in urine, hair, nails, and skin as well as in feces and sweat. The methylated forms are of lower toxicity: LD50 (mouse) values are: iAs(III) = 4.5, iAs(V) = 14–18, MMA = 1800 and DMA = 1200 mg/kg. Urinary elimination half times range from one to four days and appear to be dependent upon dose and chemical form [26,49].

## 10. Discussion

In spite of the drawbacks mentioned above [3,4], the use of hair analysis in toxicological, clinical, environmental, and forensic investigations on arsenic is growing and becoming more extensive [6,50,51,52]. The hair–urine correlation (r = 0.64, *p* < 0.001) reported by Concha et al. [11] supports Flesch’s [2] suggestion that hair could be a minor excretory organ for arsenic. The separation, identification, and quantification of arsenic and its metabolites in hair has confirmed, at least qualitatively, that arsenic is absorbed and metabolized, and a fraction of it is excreted in the hair. For the most part, HPLC–ICPMS has been employed for this confirmation. This technique requires extraction of the arsenic and arsenic metabolites from the hair. Using a pressurized hot water extraction, Piñeiro et al. [35] reported that the sum of the arsenic in the species they recovered from the Chinese reference hair GBW 07601 was 238.6 ± 5.3 p.p.b. The reference value for this material was 0.28 ± 0.05 p.p.m. arsenic [53], indicating an 85% recovery. On the other hand, data presented by Shraim et al. [30] indicated that only 39% of the arsenic present in the Japanese standard NIES No. 13 was recovered in their extraction procedure. The sample preparation methodologies have not yet been shown to quantitatively recover, from human hair, the arsenic species in their in vivo chemical forms. From time to time, national and international agencies have prepared reference materials for human hair. The total arsenic concentrations in these materials could reflect the “normal values” for the regions from which the hair was collected; i.e., Beijing 0.28 p.p.b., [53] Tokyo and Tsukuba 0.10 p.p.m. [54], Geel 0.044 p.p.m. [55]. The reference range for arsenic in hair at the Mayo Clinic Laboratory [56] is 0.0 to 0.9 p.p.m. The tabulation of concentrations of arsenic in human hair compiled by Iyengar et al. [57] contains 16 values ranging from 0.13 to 3.71 p.p.m. Eleven of the 16 values are below 1 p.p.m. There is no agreement on a well-defined, international standard for the “normal” level of arsenic species in human hair. More likely, with leadership and encouragement of an agency such as WHO, the development of standardized procedures for collection of the hair samples and extraction of the arsenic species coupled with standardized HPLC–ICPMS methods for speciation and quantification are within the realm of possibility. Rigorous programs of quality assurance/quality control will be needed for data validation. Well-defined hair reference materials in terms of iAs(III), iAs(V), MMA and DMA will be needed for use in the QA/QC protocols.

## 11. Conclusions

The utilization of hair analysis for purposes such as evaluating arsenic intoxication should be approached cautiously. There is little doubt that hair arsenic levels above 1 or 2 p.p.m. are indicative of arsenic exposure, and the presence of MMA and DMA in the hair reflects arsenic absorption. However, clinical signs and symptoms are necessary to complete the diagnosis of arsenic poisoning.

## Figures and Tables

**Table 1 ijerph-16-00977-t001:** Arsenic concentrations in blood, urine, and hair from inhabitants of the study population [9].

	Blood, p.p.b.	Urine, p.p.b.	Hair, p.p.b.	Nail, p.p.b.
Exposed	8 ± 5	300 ± 180	1240 ± 610	4550 ± 3250
Control	2 ± 1	10 ± 10	60 ± 20	420 ± 240

**Table 2 ijerph-16-00977-t002:** Average concentrations of arsenic in drinking water and urine [11].

Location	Water, p.p.b.	Urine, p.p.b.
Rosario de Lerma	0.6	7.2
Tolar Grande	2.5	15
Joaquin V. Gonzalez	6.4	10
Olacapato	14	26
Santa Rosa de los Pastos Grande	31	55
Anta	187	141
San Antonio de los Cobres	189	265
Taco Pozo	215	366

**Table 3 ijerph-16-00977-t003:** Total arsenic concentrations in organs after a suicide [25].

Organ	As, p.p.m.	Organ	As, p.p.m.
liver	147	pancreas	11.2
kidneys	26.8	lungs	11.1
muscle	12.3	cerebellum	11
heart	11.8	brain	8.8
spleen	11.7	skin	2.9

**Table 4 ijerph-16-00977-t004:** Distribution of arsenic species in organs, % of total arsenic [25].

Organ	As (III)	As (V)	DMA	MMA
liver	83	2	4	10
kidneys	75	2	6	17
muscle	75	<1	6	16
heart	77	<1	5	14
spleen	81	<1	5	13
pancreas	82	<1	4	10
lung	85	<1	4	10
cerebellum	47	<1	19	30
brain	53	<1	18	27
skin	56	<1	15	28

MMA = monomethylarsonic acid; DMA = dimethylarsinic acid.

**Table 5 ijerph-16-00977-t005:** Possible arsenic species for methylation and elimination.

Compound	Molecular Formula	Arsenic Oxidation State
arsenous acid	As(OH)_3_	+3
arsenic acid	AsO(OH)_3_	+5
monomethylarsonous acid	(CH_3_)As(OH)_2_	+3
monomethylarsonic acid	(CH_3_)AsO(OH)_2_	+5
dimethylarsinous acid	(CH_3_)_2_As(OH)	+3
dimethylarsinic acid	(CH_3_)_2_AsO(OH)	+5

**Table 6 ijerph-16-00977-t006:** Arsenic speciation in hair from poisoning survivors and in NIES CRM No. 13, p.p.b. [30].

Sample	As (III) *	DMA	MMA	As(V) **	Sum Total
1	126	19	13	435	593
2	78	39	21	568	708
CRM	6	3	4	26	39 ***

* As (III) is inorganic trivalent arsenic.; ** As(V) is inorganic pentavalent arsenic. *** The reference value for total arsenic in NIES CRM No. 13 is 100 p.p.b. [31].

**Table 7 ijerph-16-00977-t007:** Summary of arsenic species extracted from hair of APL * patients [18].

Species	Mean, p.p.b.	Range p.p.b.
As (III)	900.9	5.1–8059.5
As (V)	90.1	1.9–895.0
MMA	9.4	<1–60.7
DMA	47.6	<1–350.4
monomethylarsonous acid	36.8	<1–104.4
monomethylmonothioarsonic acid	11.1	<1–20.7
dimethylmonothioarsnic acid	56.1	<1–84.0
unknown arsenic species 1	16.3	<1–28.5
unknown arsenic species 2	4.4	<1–13.8

* APL, acute promyelocytic leukemia.

**Table 8 ijerph-16-00977-t008:** Relative concentrations of As(III), As(V), MMA and DMA normalized to DMA.

Sample	Reference	As (III)	As (V)	MMA	DMA
hair	[29]	6.6	22.9	0.68	1
hair	[29]	2	17.6	0.54	1
hair	[31]	16.9	9.2	0.61	1
hair	[18]	18.9	1.9	0.2	1
hair	[38]	74	158	<LoD	1
hair	[38]	298	355	<LoD	1
hair	[39]	17.3	10.2	0.85	1
urine *	[40]	0.21	0.01	0.3	1
urine	[41]	total iAs = 0.30	0.27	1	
urine	[42]	total iAs = 0.16	0.21	1	
urine	[43]	0.16	0.21	0.31	1

* Industrial accident with arsine gas.

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
