# Peer review of "On the Use of Hair Analysis for Assessing Arsenic Intoxication"

_ijerph, 2019, doi:10.3390/ijerph16060977_

Round 1

Reviewer 1 Report

This interesting review covers a large number of studies dealing with hair analysis for arsenic.  Several points could use attention.

[1] Many quotations are used, some of which seem unnecessary or redundant and raise questions.  For example, in the long quotation from Katz and Wood [3] (lines 59-73), is the author endorsing the idea in point 1 that “normal values showing narrow ranges” are needed?  The need for narrow ranges is unclear if exposures are widely variant.  Overall, the text could be shortened.

[2] Section 4 on application and mis-application drags the reader through a history of measurements of hair from Napolean Bonaparte for reasons that are not well articulated.  This section could use condensation and more clear indication of which measurements were mis-applications.

[3] The phrase that at least 6 forms “may be involved” in arsenic methylation is vague and confusing (referring to Table 5).  A more illuminating explanation could be given, such as sequential methylation of a reduced species (III) that is then oxidized (V) as described in the work of D. J. Thomas and co-workers. 

[4] Also regarding Table 5, line 306 states that these species are of metabolic origin, but arsenic acid is not.

[5] The sentence in lines 328-329 referring to “a relationship” is so vague that its intended meaning is obscure.

[6] The text refers to endogenous incorporation of arsenic into the hair as absorption, but this is easily confused with adsorption due to exogenous exposure.

[7] The text would benefit from careful editing for minor errors.  The funding and acknowledgments sections need completion.

Author Response

REVIEWER 1:

This interesting review covers a large number of studies dealing with hair analysis for arsenic. Several points could use attention.

[1] Many quotations are used,some of which seem unnecessary or redundant and raise questions. For example, in the long quotation from Katz and Wood [3] (lines 59-73), is the author endorsing the idea in point 1 that “normal values showing narrow ranges” are needed? The need for narrow ranges is unclear if exposures are widely variant.

The use of direct quotations avoid the possibilities for the author's misinterpretations of the work being cited. The quotation in lines 59–73 shows the objections to hair mineral analysis have persisted for decades have not yet been resolved completely.

In the human, an oral temperature of 97 to 99 degrees Fahrenheit is "normal", and a blood glucose between 70 and 130 mg/dL is "normal". The establishment of "normal" values is important for the assessing health or illness. The establishment of a "normal" value for arsenic is of the same importance. This is obvious and requires no further elaboration.

Overall, the text could be shortened.

Yes, the text could be shortened, but why? Is there a limit on the number of words or lines or pages?

[2] Section 4 on application and miss-application drags the reader through a history of measurements of hair from Napoleon Bonaparte for reasons that are not well articulated. This section could use condensation and more clear indication of which measurements were misapplications.

None of the measurements are misapplications. The interpretation of the results for the measurement of the arsenic concentrations in Napoleon's hair demonstrates the conflict and controversy associated with the usefulness of hair mineral analysis.

Rather than condense this section, the author suggests it be eliminated along with references 7 through 21. Also such elimination may satisfy the reviewer's comment, "Overall, the text could be Shortened." The work of Peter Flesch can be included in the introduction.

[3] The phrase that at least 6 forms “may be involved” in arsenic methylation is vague and confusing (referring to Table 5). A more illuminating explanation could be given, such as sequential methylation of a reduced species (III) that is then oxidized (V) as described in the work of D. J. Thomas and coworkers.

The conceptual models of Thomas, et al. (Exp. Biol. Med., 2007 232(1), 3-13) can be described in a versed manuscript.

[4] Also regarding Table 5, line 306 states that these species are of metabolic origin, but arsenic acid is not.

The inorganic arsenic species, arsenous acid, H3AsO3, and arsenic acid, H3AsO4, are not metabolic products. Their origins are external to the metabolic environment.

[5] The sentence in lines 328-329 referring to “a relationship” is so vague that its intended meaning is obscure.

A modifier such as mathematical can be inserted, or the word, relationship, can be replaced with correlation.

[6] The text refers to endogenous incorporation of arsenic into the hair as absorption, but this is easily confused with adsorption due to exogenous exposure.

The parenthetical inclusion of absorption and adsorption may resolve this issue; endogenous (absorption) and exogenous (adsorption).

[7] The text would benefit from careful editing for minor errors.

The manuscript will be revised and proof read.

The funding and acknowledgments sections need completion.

No funding was received for the preparation of this manuscript. No acknowledgments are necessary.

Reviewer 2 Report

This is a review article and fits perfectly into the scopes of the journal. The author briefly mentioned why and what write, and strictly followed this throughout the document. Each written fragment fully corresponds to its title. Concise sentences, huge and very difficult, complex matter that has been successfully condensed to a small area. It is obvious that the author is an expert in his knowledge of arsenic toxicology and analysis. Minor technical errors and inaccuracies are mentioned in the comments in the attached PDF file.

I suggest changing the type of the Paper from “Article” to “Review”.

Author Response

REVIEWER 2:

This is a review article and fits perfectly into the scopes of the journal. The author briefly mentioned

why and what write, and strictly followed this throughout the document. Each written fragment fully

corresponds to its title. Concise sentences, huge and very difficult, complex matter that has been

successfully condensed to a small area. It is obvious that the author is an expert in his knowledge of arsenic toxicology and analysis. Minor technical errors and inaccuracies are mentioned in the

comments in the attached PDF file.

I suggest changing the type of the Paper from “Article” to “Review”.

This manuscript is indeed a review. The author has no objections to identifying it as such.
